# Active Components of *Leontopodium alpinum* Callus Culture Extract for Blue Light Damage in Human Foreskin Fibroblasts

**DOI:** 10.3390/molecules28217319

**Published:** 2023-10-28

**Authors:** Haodong Li, Xianyao Meng, Ying Zhang, Miaomiao Guo, Li Li

**Affiliations:** College of Chemistry and Materials Engineering, Beijing Technology & Business University, Beijing 100048, China; lhdong0625@163.com (H.L.); mengxy8896@163.com (X.M.); 13126598550@163.com (Y.Z.); guomiaomiao7@163.com (M.G.)

**Keywords:** *Leontopodium alpinum*, OPN3, calcium influx, ROS, leontopodic acid A

## Abstract

*Leontopodium alpinum* is a source of raw material for food additives and skin health. The purpose of this study was to investigate the application of *Leontopodium alpinum* callus culture extract (LACCE) to prevent blue light damage to the skin. We screened and identified the blue light-damage-protecting activities and mechanisms of ten components of LACCE, including chlorogenic acid (A), isoquercitrin (B), isochlorogenic acid A (C), cynaroside (D), syringin (E), isochlorogenic acid (F), cynarin (G), rutin (H), leontopodic acid A (I), and leontopodic acid B (J), using a novel blue light-induced human foreskin fibroblast (HFF-1) cell injury model. The study examined the cytotoxicity of ten ingredients using the cell counting kit-8 (CCK-8) assay, and selecting concentrations of 5, 10, and 20 μM for experiments with a cell viability above 65%. We explored the effects and mechanisms of action of these LACCE components in response to blue light injury using Western blotting and an enzyme-linked immunosorbent assay. We also measured ROS secretion and Ca^2+^ influx. Our study revealed that leontopodic acid A effectively boosted COI-1 expression, hindered MMP-1 expression, curbed ROS and Ca^2+^ endocytosis, and reduced OPN3 expression. These results provide theoretical support for the development of new raw materials for the pharmaceutical and skincare industries.

## 1. Introduction

*Leontopodium alpinum*, commonly known as edelweiss, is a perennial solar herb that grows up to 20 cm in height [1]. *L. alpinum* is the national flower of Switzerland and Austria, native to the Alpine Mountains in Europe, and grows in high-altitude limestone areas. It is usually located between 1000 and 3400 m above sea level [2]. Due to its harsh growing environment, growing human demand, and scarce resources, *L. alpinum* is now classified as a rare species. Nonetheless, it exhibits resistance to severe weather and possesses a significant self-healing ability. Moreover, it showcases distinctive pharmacological properties, including antibacterial [3], antioxidant [4], lipid regulation [5], anti-inflammatory [6], and analgesic effects [7]. However, *L. alpinum* is now considered a rare species due to its harsh growing environment, growing human demand, and scarce resources.

Several classes of secondary metabolites, including terpenoids, phenylpropane derivatives, and aliphatic compounds, have been isolated from various parts of *L. alpinum* for some 50 years [1]. Studies have demonstrated that leontopodic acids exhibit properties that effectively combat inflammation and oxidation [8]. Schwaiger et al. established the antioxidant potency of leontopodic acids using a range of in vitro techniques, including the Briggs–Rauscher oscillating reaction (BR) method and Trolox equivalent antioxidant capacity (TEAC) assay [9]. The results showed that leontopodic acids exhibit remarkable scavenging ability against free radicals, with up to four times greater activity than resorcinol in the BR assay and twice the efficacy of Trolox in the TEAC assay [9]. Costa et al. demonstrated that leontopodic acid (Figure 1) significantly reduced ROS formation [3]. Cho et al. examined the in vivo effects of *Leontopodium alpinum* callus culture extract (LACCE). They clinically evaluated LACCE based on different factors, including facial lifting and the improvement of periorbital wrinkles, skin elasticity, dermal density, and skin thickness. The number of patients with decreased periorbital wrinkle roughness was significantly higher in the LACCE group than in the placebo group. The growth rates of skin elasticity, density, and thickness were significantly higher in the LACCE group than in the placebo group. This indicates that LACCE can inhibit wrinkles and increase moisturizing activity [10]. Meng et al. investigated the effect of LACCE on blue light injury and found that LACCE promoted COI-1 production, inhibited the secretion of MMP-1 and ROS, and inhibited inward Ca^2+^ flow. In addition, LACCE significantly inhibited expression of OPN3, suggesting that it may reduce activation of the OPN3 calcium pathway by blue light [11]. Based on the results of these studies, there is merit in further investigating the antioxidant properties and potential anti-aging effects of leontopodic acid.

Blue light has the shortest wavelength and highest energy in the visible light wavelength band, typically ranging from 420 to 490 nm. LED lights in computer screens, cell phones, and televisions produce large amounts of blue light that can harm the eyes and skin [12]. Visible light, specifically high-intensity blue light, can penetrate the skin deeper than UVB and UVA rays. Recent scientific studies indicate that blue light can cause permanent damage to skin cells in the dermis and epidermis. Consequently, this damage can affect the DNA within these cells and lead to the deterioration of support fibers in the skin. As a result, long-term consequences may include the appearance of fine lines, wrinkles, and premature aging [13].

In recent years, many plant extracts have been investigated for their ability to prevent blue light-induced premature photoaging of the skin. Jun et al. found that a *Prunella vulgaris* extract exerted protective effects against blue light-induced oxidative stress and inflammation through an early activation of the Nrf-2/HO-1 pathway and later inhibition of NF-kB translocation [14]. Jong et al. showed the high blue light absorption and antioxidant activity of *Caesalpinia sappan* extract [15]. Rosanna et al. used hydroxyloctanol extracted from olives, and found that it protected keratinocytes and fibroblasts from blue-light-induced damage [16]. Irene et al. discovered that the expression of OPN3 increased in response to blue light irradiation, leading to the stimulation of skin wound closure. Using gene-silencing techniques, they could identify OPN3 as the receptor responsible for sensing blue light. This receptor can regulate keratinocyte differentiation and restore the skin barrier function [17].

Opsins (OPNs) are a class of photoreceptors that have recently attracted considerable attention. The OPN family, part of the photosensitive G protein-coupled receptor (GPCR) superfamily, is involved in phototransduction through the GPCR signaling pathway [18,19,20,21,22]. In humans, the OPN family consists of five subfamilies: OPN1 (cone opsin), OPN2 (rhodopsin), OPN3 (encephalopsin and tmt-opsin), OPN4 (melanopsin), and OPN5 (neuropsins) [21]. OPN3 is a non-visual OPN that is highly expressed in the skin, and has been shown to upregulate tyrosinase activity in human epidermal melanocytes [23]. OPNs are chromophores responsible for the direct effects of blue light and their activation leads to an overproduction of ROS [24]. Lan et al. demonstrated that OPN3 functions as a UVA phototransduction sensor in fibroblasts, playing a role in the upregulation of MMP-1 and MMP-3 expression [25]. They found that blue light exposure increased calcium flux and the upregulation of pCAMKII expression, a vital enzyme in the calcium pathway. In addition, the downregulation of OPN3 expression blocked blue-light-induced calcium flux and phosphorylation of calcium/CaM-dependent protein kinase II (CAMKII), CREB, extracellular regulating kinases (ERK), and p38 [25]. 

In a previous article, we suggested that LACCE may produce anti-blue light damage by inhibiting activation of the OPN3/Ca^2+^-dependent signaling pathway and the onset of ROS oxidative stress, thereby inhibiting MMP-1 secretion and promoting COI-1 synthesis [11]. Leontopodic acid A (I) is a constituent specific to snowdrops and is structurally a derivative of glucaric acid substituted with caffeoyl and 3-hydroxybutanyl moieties [26]. However, few studies have been conducted on leontopodic acid A monomers. Schwaiger et al. [9] showed that leontopodic acid has a potent free radical scavenging ability in alpine flamingos, as well as potent antioxidant activity in DNA assays. In U937 cells, leontopodic acid increased glutathione peroxidase (GPX) [27]. GPX catalyzes a peroxidation reaction involving reduced glutathione, which scavenges peroxides and hydroxyl radicals generated during cellular respiratory metabolism, thereby reducing antioxidant activity due to peroxidation of polyunsaturated fatty acids in the cell membrane. Finally, Yi et al. [28] found that leontopodic acid A (I) inhibited the accumulation of triglycerides in HepG 2 cells.

Our previous study showed that healing tissue using plant cell extracts from *L. alpinum* can protect against blue light damage, and that this effect may be achieved by acting on the OPN3/Ca^2+^-dependent signaling pathway [11]. In our previous study, the *L. alpinum* callus culture was provided by Ancelbio Biotechnology Co., Ltd. (Chongqing, China) [11]. On the other hand, the active ingredients in LACCE were quantified using high-performance liquid chromatography (HPLC) and ultra-performance liquid chromatography–tandem mass spectrometry (UPLC–MS/MS) [11]. In order to investigate the role of the active components of LACCE against blue light damage, this study used a novel blue light-induced human foreskin fibroblast (HFF-1) cell injury model. We investigated the efficacy of 10 commercially available active ingredients from *L. alpinum* in protecting against blue light damage, as well as their effects on blue light-induced secretion of COL-1 and MMP-1 in HFF using an ELISA and Western blotting, and ROS and Ca^2+^ using flow cytometry. We also examined the effect of leontopodic acid A (I) on OPN3 expression via Western blotting.

## 2. Results

### 2.1. Effects of Various Concentrations of 10 Active Ingredients on HFF Cell Viability

We used the CCK-8 assay to measure the effect of different doses of blue light irradiation on the viability of HFF. Figure 2 shows that the survival rate of HFF cells was above 65% at 5, 10, and 20 μM concentrations of rutin (H). All the other active ingredients had a cell survival rate of 75% or more. We chose three concentrations (5, 10, and 20 μM) for the next experiment.

### 2.2. Effect of 10 Active Ingredients on COL-Ⅰ Levels

The ELISA and Western blotting results in Figure 3 show that compared with the control group (CG), COL-I contents were significantly decreased in the blue light model group (*p* < 0.01). Compared with the model group (MC), all 10 active ingredients promoted COL-Ⅰ secretion in HFF cells, among which chlorogenic acid (A), isoquercitrin (B), isochlorogenic acid A (C), cynarin (G), and rutin (H) showed significant effects on COL-Ⅰ content at 5, 10, and 20 μM concentrations.

The experimental results for the two components of *L. alpinum* that have received much attention, leontopodic acid A (I) and leontopodic acid B (J), showed that leontopodic acid A (I) could significantly increase COI-1 content in HFF cells above that of the control group (CG).

### 2.3. Effects of the 10 Active Ingredients on MMP-1 Levels

MMP-1 levels were significantly higher in the blue light model group (MC) than in the control group (CG) (*p* < 0.01), as shown in Figure 4. Each of the 10 active ingredients decreased MMP-1 secretion in HFF cells at 10 μM compared with the blue light model group, in which all groups, i.e., chlorogenic acid (A), isoquercitrin (B), isochlorogenic acid A (C), isochlorogenic acid (F), cynarin (G), and rutin (H), significantly increased the secretion of MMP-1. Group I/J significantly decreased MMP-1 content at 20 μM.

### 2.4. Effects of LACCE on ROS Levels in HFF Cells

Compared with the control group (CG), ROS levels were significantly reduced in the blue light model group (*p* < 0.01), as shown in Figure 5. Each of the 10 active ingredients reduced the ROS contents of HFF to some extent compared to those of the blue light model group. Among them, leontopodic acid A (I) had the strongest effect on ROS scavenging activity.

### 2.5. Effects of LACCE on Ca^2+^ Inflows in HFF Cells

Calcium ion inflow was measured via flow cytometry (Figure 6). Compared with the blank control group (CG), the Ca^2+^ inflow in the blue light model group (MC) increased significantly, and the difference was statistically significant (*p* < 0.01). Compared with the effects of the ingredients on the blue light model group, all 10 active ingredients reduced calcium ion influx, with the most pronounced effect observed in group H.

### 2.6. Effect of Leontopodic Acid A (I) on OPN Expression in HFF Cells

OPN3 expression was higher in the blue light model group (MC) than in the control group (CG) (*p* < 0.01), as shown in Figure 7. Compared with the OPN expression in the blue light model group, leontopodic acid A (I) reduced the expression of OPNs at 5, 10, and 20 μM, with the most pronounced effect at 20 μM, which was an even lower concentration than that applied in the control group (CG).

## 3. Discussion

The overproduction of ROS, which can be induced by blue light, is associated with reduced cell viability and proliferation, increased pro-inflammatory signaling, and increased collagen metabolic terminals. Dermal fibroblasts exposed to blue light result in a comprehensive inhibition of metabolism, compromised signaling of transforming growth factor-β (TGF-β), decreased synthesis of adenosine triphosphate (ATP), and reduced procollagen I synthesis. As shown in this study, in addition to ROS overproduction, blue light exposure also increases MMP-1 release.

Blue light causes skin pigmentation. Although both UV and blue light cause hyperpigmentation via ROS production, only blue light causes hyperpigmentation via opsin stimulation. OPN3 triggers a signaling cascade involving the calcium-dependent activation of calcium/calmodulin-dependent protein kinase II (CAMKII) in response to blue light. This is followed by the activation of cyclic adenosine monophosphate (cAMP) response element binding protein (CREB), extracellular signal-regulated kinase (ERK), and p38, ultimately leading to the phosphorylation of microphthalmia-associated transcription factor (MITF) and subsequent activation [29]. Recent reports have confirmed that OPN3 is the critical sensor responsible for hyperpigmentation induced by blue light [30]. The knockdown of OPN3 after blue light exposure did not affect the DNA synthesis rate of keratin-forming cells, but resulted in reduced early differentiation of keratin-forming cells after blue light exposure [17], suggesting that OPN3 is necessary for skin barrier restoration.

Our previous studies established that LACCE can promote COI-1 production, inhibit MMP-1 secretion, and reduce blue light damage [11]. To further investigate the anti-blue light damage and mechanism of action of LACCE, we examined the effects of the 10 main active components of LACCE using a blue light-induced HFF cell injury model.

The study of safe concentrations of active ingredients in HFF cells found that except for cynaroside (D), the active ingredients may be toxic to cells at increased concentrations. Among them, the active ingredient rutin (H) showed the highest toxicity to the cells when the concentration reached 20 μM. However, since the toxicity of the other active ingredients to cells was greater than 75% at the concentrations of 5, 10, and 20 μM, we chose these three concentrations for subsequent studies. Notably, increasing concentrations of cynaroside (D) increased cell survival, which may indicate that elevated concentrations of cynaroside (D) promote cell proliferation within a specific concentration range.

The Western blotting and enzyme-linked immunosorbent assay results revealed that cynarin (G), rutin (H), and leontopodic acid A (I) at 20 μM significantly promoted COI-1 expression. In contrast, chlorogenic acid (A), isoquercitrin (B), isochlorogenic acid A (C), syringin (E), cynarin (G), and rutin (H) significantly decreased MMP-1 expression at 10 μM. Similarly, leontopodic acid A (I) and leontopodic acid B (J) significantly reduced MMP-1 expression at 20 μM. Furthermore, the study of ROS levels and Ca^2+^ influx showed that the effects of chlorogenic acid (A) and leontopodic acid A (I) on ROS scavenging were significant. Compared to the effects in the model group, all 10 active ingredients reduced Ca^2+^ influx to different degrees. These results suggest that the 10 active ingredients demonstrated the regulation of COI-I and MMP-1 at specific concentrations and the inhibitory effect on scavenging ROS and Ca^2+^ influx at different concentrations. The effect of LACCE on the OPN3/Ca^2+^-dependent signaling pathway has been confirmed in previous studies [11], and experiments with these 10 active ingredients have also laterally verified that LACCE has an anti-blue light damage or protective effect. 

Through our previous studies on the composition of LACCE, leontopodic acid A (I) and leontopodic acid B (J) are determined to be the main components of LACCE, among which leontopodic acid A (I) makes up the highest content [11]. Therefore, we further investigated the effect of leontopodic acid A (I) on OPN3 and found that, by enhancing the production of COI-1 while reducing the release of MMP-1, leontopodic acid A (I) showed promising results. Additionally, leontopodic acid A (I) has been observed to be effective in scavenging ROS and hindering Ca^2+^ influx. The data from the model group showed that OPN3-treated cells responded to blue light exposure. Moreover, leontopodic acid A (I) significantly reduced OPN3 at 20 μM, suggesting that leontopodic acid A (I) has a significant effect on the OPN3 pathway. 

In summary, the anti-blue light effect of LACCE was further confirmed by the study of the ten significant components of LACCE, the effect of which may be produced through the OPN3/Ca^2+^-dependent signaling pathway. The discovery of raw materials that target OPN-mediated signaling pathways opens new frontiers for the treatment of skin photoaging. Further studies are required to confirm the responses of OPN3 to UVR and blue light. 

In addition, this study was performed on cells rather than using an intact skin model. Skin states are complex, and the observed results may differ depending on the model used. In addition, the effects of sunlight on skin are different from those of blue light emitted by electronic devices. Therefore, the protective effects of LACCE against blue light must be further refined and studied using a skin model.

## 4. Materials and Methods

### 4.1. Cell Culture and Materials

A Trypsin EDTA Solution A (0.25% Trypsin -EDTA 0.02% in HBSS) (BIOAGRIO, Mountain View, CA, USA) and A COL-I/MMP-1 (ELISA) kit (Biotech, Shanghai, China) were employed. The primary antibody against COL-I/MMP-1 was used (Abcam, Cambridge, UK). Additionally, chlorogenic acid (A), isoquercitrin (B), isochlorogenic acid A (C), cymaroside (D), syringin (E), isochlorogenic acid (F), cynarin (G), rutin (H), leontopodic acid A (I), and leontopodic acid B (J) were used in the study (Yuanye Biotechnology Co., Ltd., Shanghai, China).

### 4.2. Cell Culture

Human foreskin fibroblast (HFF) cells (The National Laboratory Cell Resource Sharing Platform, Beijing, China) were cultured in Dulbecco’s Modified Eagle’s Medium (DMEM) (Wisent Biotechnology (Nanjing) Co., Ltd., Nanjing, China) containing 15% heat-inactivated fetal bovine serum (FBS) (BIOAGRIO, Mountain View, CA, USA) and 1% penicillin–streptomycin (PS) (Gibco, Thermo Fisher Scientific, Waltham, MA, USA) at 37 °C and 5% CO_2_. The cultured cells were used for experiments when they were in the logarithmic growth phase.

### 4.3. Cell Viability Assay

The study used a cell counting kit-8 assay (CCK-8) (Biotechwell, Shanghai, China) to determine cellular growth, propagation, and survival. Briefly, HFF cells were cultured in 96-well plates at a density of 1 × 10^4^ cells/well for 12 h (37 °C and 5% CO_2_). Three parallel wells were set up for each group. Then, the HFF cells were pretreated with various concentrations (5, 10, 20 μM) of 10 active ingredients. The experimental groups corresponding to the active ingredients are listed in Table 1. After 24 h (37 °C and 5% CO_2_) of incubation, the CCK-8 solution was added to the cells for 1 h at 37 °C under light-free conditions. Subsequently, the absorbance of the samples was measured at 450 nm using a microplate reader (TECAN, Männedorf, Switzerland) to calculate the cell viability. 

### 4.4. Measurement of COL-I, MMP-1, and OPN3 Levels Using ELISA

HFF cells at the logarithmic growth stage were seeded in 6-well plates at a density of 3 × 10^5^ cells/well, with 2 mL of medium per well, and incubated at 37 °C with 5% CO_2_ for 12 h. Three groups were set up: a blank group, a blue light model group, and a sample group. The blank and blue light model groups were supplemented with 2 mL of a serum-free medium, while the sample group received 2 mL of a medium containing different concentrations (5, 10, 20 μM) of 10 active ingredients. After 6 h of further incubation at 37 °C with 5% CO_2_, the cells were exposed to blue light irradiation. Subsequently, 2 mL of a serum-free medium was added to each well, and the cells were cultured in an incubator at 37 °C with 5% CO_2_ for 18 h. The levels of COL-I and MMP-1 were measured using a COL-I/MMP-1/OPN3 ELISA kit, following the manufacturer’s instructions. 

### 4.5. Measurement of COL-I, MMP-1, and OPN3 Levels Using Western Blot

The cell culture, and blue light radiation are described in Section 4.4. Following incubation with blue light radiation, the cells were gathered, and their total protein was extracted using a total protein lysis solution. This extracted protein was then centrifuged at 12,000× *g* for 10 min, resulting in the extraction of the supernatant. To determine protein concentrations, a BCA protein detection kit was utilized. The protein denaturation process involved pre-heating the metal bath at 100 °C for 5 min. The protein samples were then subjected to 10% SDS-PAGE electrophoresis and transferred onto a PVDF membrane, which was incubated with 5% BSA at room temperature for 1 h to block the membrane. Following three washes with TBST, the membrane was placed in a solution containing primary antibodies β-actin and COL-Ⅰ/MMP-1/OPN3 (Abcam, Cambridge, UK), and left overnight at 4 °C. After three further washes with TBST, the membrane was incubated with the corresponding secondary antibody at room temperature for 1 h. Finally, the protein bands were visualized using ECL reagents and a chemiluminescent detection system.

### 4.6. Flow Cytometry Analysis of ROS Production Analysis

The cells were rinsed with PBS and then treated with 1 mL of 10 μM DCFH-DA. Next, the cells were placed in a dark 37 °C incubator for 30 min. After that, the DCFH-DA was removed, and the cells were rewashed with PBS and diluted to a concentration of 1 × 10^5^ cells/mL. The cells were analyzed using the flow cytometer FL2 channel.

### 4.7. Flow Cytometry Analysis of Ca^2+^ Production

An amount of 10 μL of a pre-configured working solution of Fluo-4 AM (4 μM) was added to the cells, and they were then incubated for 20 min at 37 °C. After removing the Fluo-4 AM working solution, the cells were washed three times following the same procedure. The AM was cultured at 37 °C for 10 min to ensure complete cell de-esterification. An appropriate amount of PBS was added to the cells to prepare a 1 × 10^5^ cells/mL cell solution. Finally, the cells were observed using the FL2 channel of a flow cytometer.

### 4.8. Statistical Analysis

The data were presented as the average ± standard error of the mean (SEM). The *t*-test was utilized to analyze the discrepancies between the two groups. For multiple-group comparisons, a one-way analysis of variance (ANOVA) was performed using IBM SPSS Statistics software (version 26.0; IBM Corp., Armonk, NY, USA). A *p*-value below 0.05 was deemed statistically significant.

## 5. Conclusions

As a plant material that has attracted much attention in recent years, LACCE and its active compounds, including chlorogenic acid (A), isoquercitrin (B), isochlorogenic acid A (C), cynaroside (D), syringin (E), isochlorogenic acid (F), cynarin (G), rutin (H), leontopodic acid A (I), and leontopodic acid B (J), deserve further study. The mechanism of action of LACCE and its active ingredients on the OPN3/Ca^2+^-dependent signaling pathway also needs to be further explored.

Here, we examined the efficacy of the 10 main active ingredients in LACCE and found that they could promote the production of COL-I, inhibit the secretion of MMP-1, reduce the level of ROS, decrease the inward flow of Ca^2+^, and mitigate blue light damage to different degrees. Leontopodic acid A (I), specifically, significantly downregulated OPN3 expression. These results suggest that the active ingredients in LACCE have anti-blue light damage effects on blue light-induced fibroblast models, providing further theoretical support for the application of LACCE in skin care, medicine, and other industries. This study provides data support for the development of functional ingredients. Future research should consider the effects of LACCE on cell signaling.

## Figures and Tables

**Figure 1 molecules-28-07319-f001:**
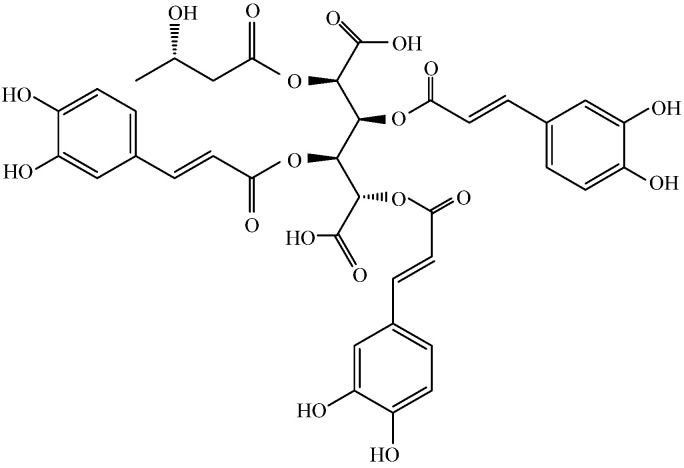
Structure of leontopodic acid A (I).

**Figure 2 molecules-28-07319-f002:**
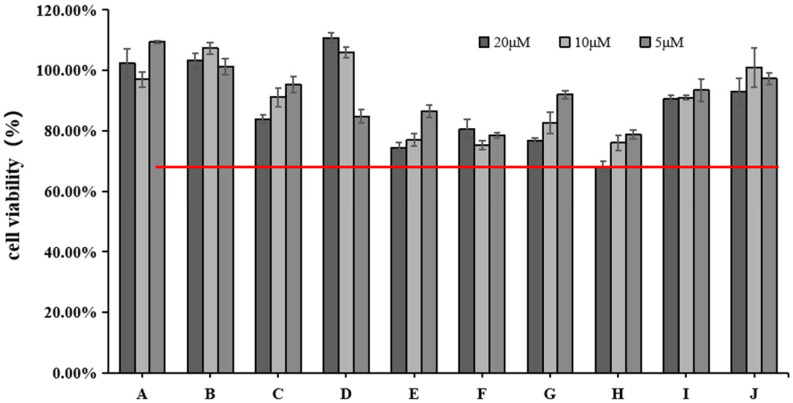
Effects of different concentrations of 10 active ingredients on HFF cell viability. The red line indicates cell viability ≥65%.

**Figure 3 molecules-28-07319-f003:**
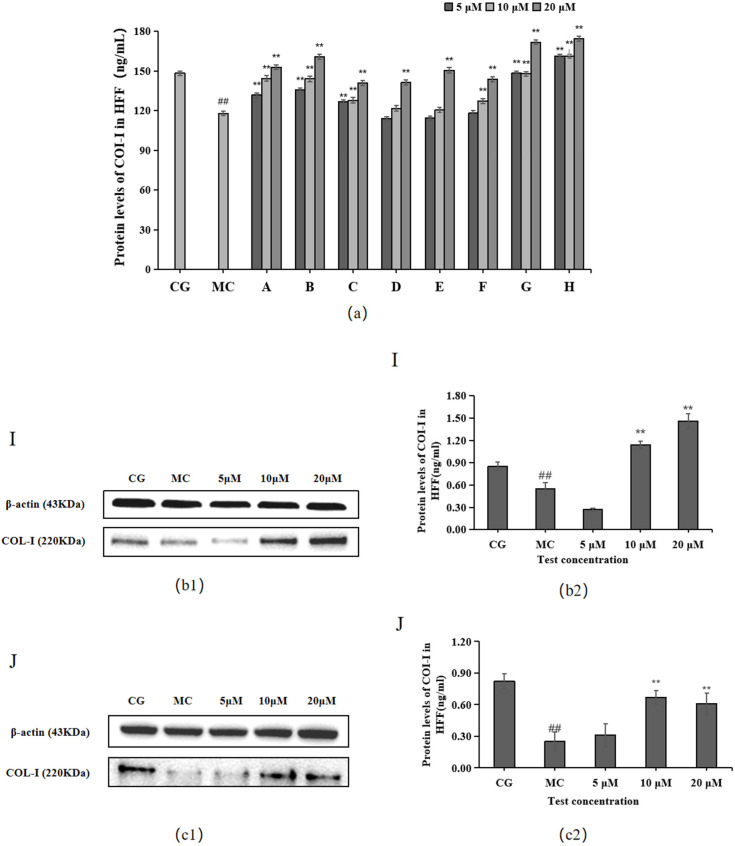
Effects of different concentrations of the 10 active ingredients on COL-I. (**a**) Effect of different concentrations of groups A–H on the expression of COI-1. (**b1**,**c1**) COI-1 expression was measured using a Western blotting assay. (**b2**,**c2**) COI-1 expression was visualized using an ELISA. The data are expressed as the mean ± standard deviation (SD) of three independent experiments (*n* = 3). ## *p* < 0.01 vs. control group (CG). ** *p* < 0.01 vs. model group (MC).

**Figure 4 molecules-28-07319-f004:**
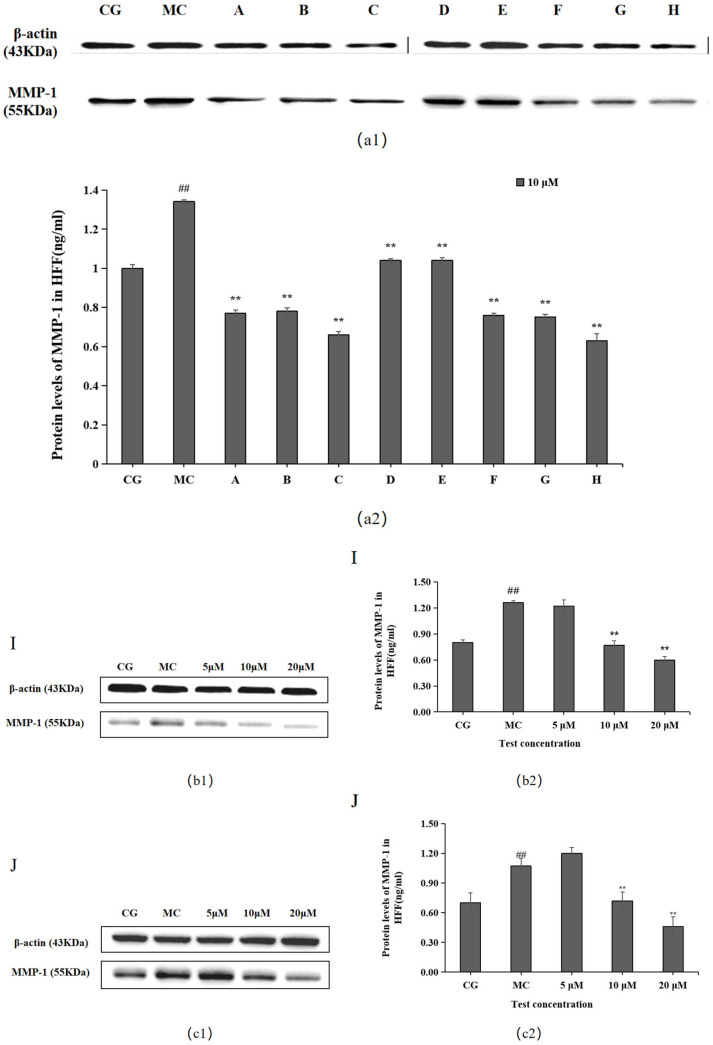
Effects of different concentrations of the 10 active ingredients on MMP-1. (**a**) Effects of different concentrations of groups A–H on MMP-1 expression: (**a1**,**b1**,**c1**) MMP-1 expression was measured using a Western blotting assay. (**a2**,**b2**,**c2**) MMP-1 expression was visualized using an ELISA. The data are expressed as the mean ± standard deviation (SD) of three independent experiments (*n* = 3). ## *p* < 0.01 vs. control group (CG). ** *p*< 0.01 vs. model group (MC).

**Figure 5 molecules-28-07319-f005:**
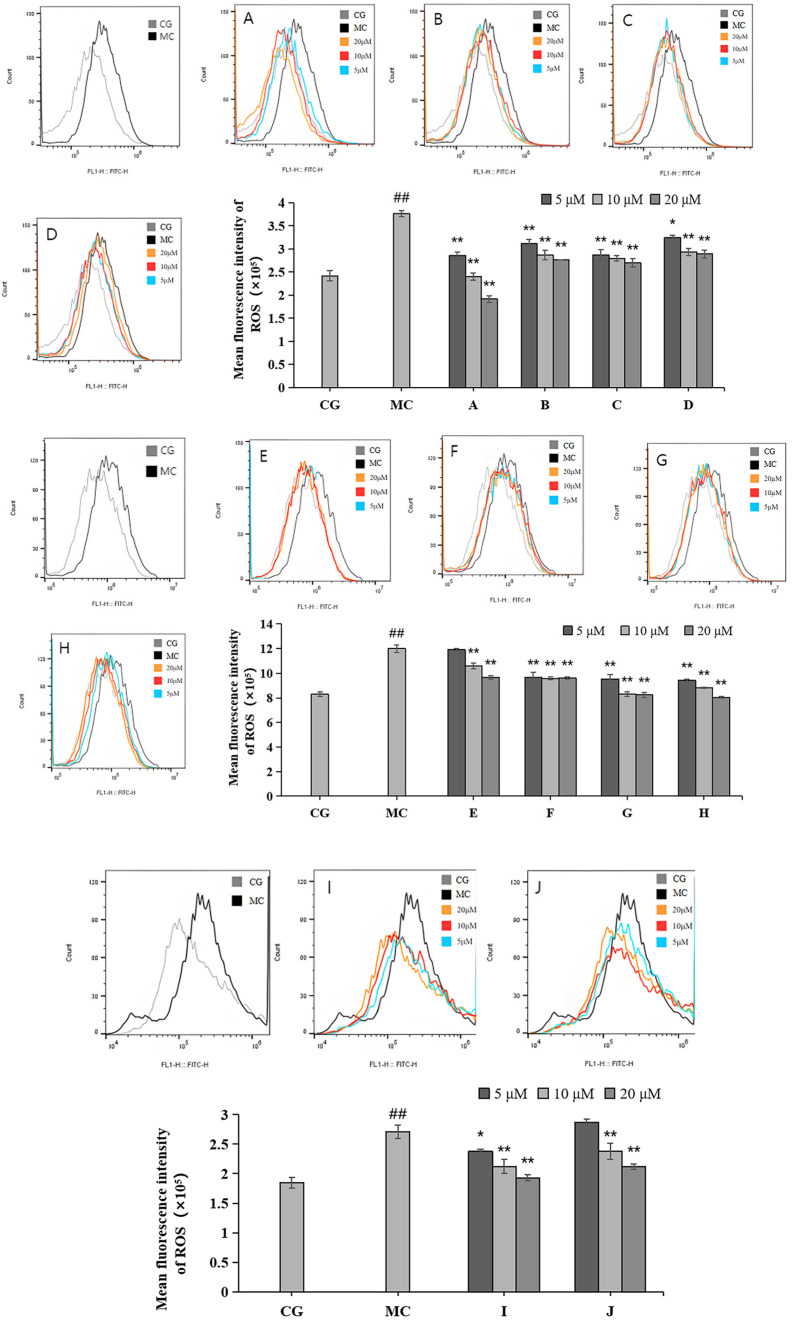
Effects of different concentrations of groups A–J on ROS levels measured via flow cytometry. Quantitative analysis of ROS fluorescence intensity (*n* = 3). The data are expressed as the mean ± standard deviation (SD) of three independent experiments. ## *p* < 0.01 vs. control group (CG); * *p* < 0.1, ** *p* < 0.01 vs. model group (MC).

**Figure 6 molecules-28-07319-f006:**
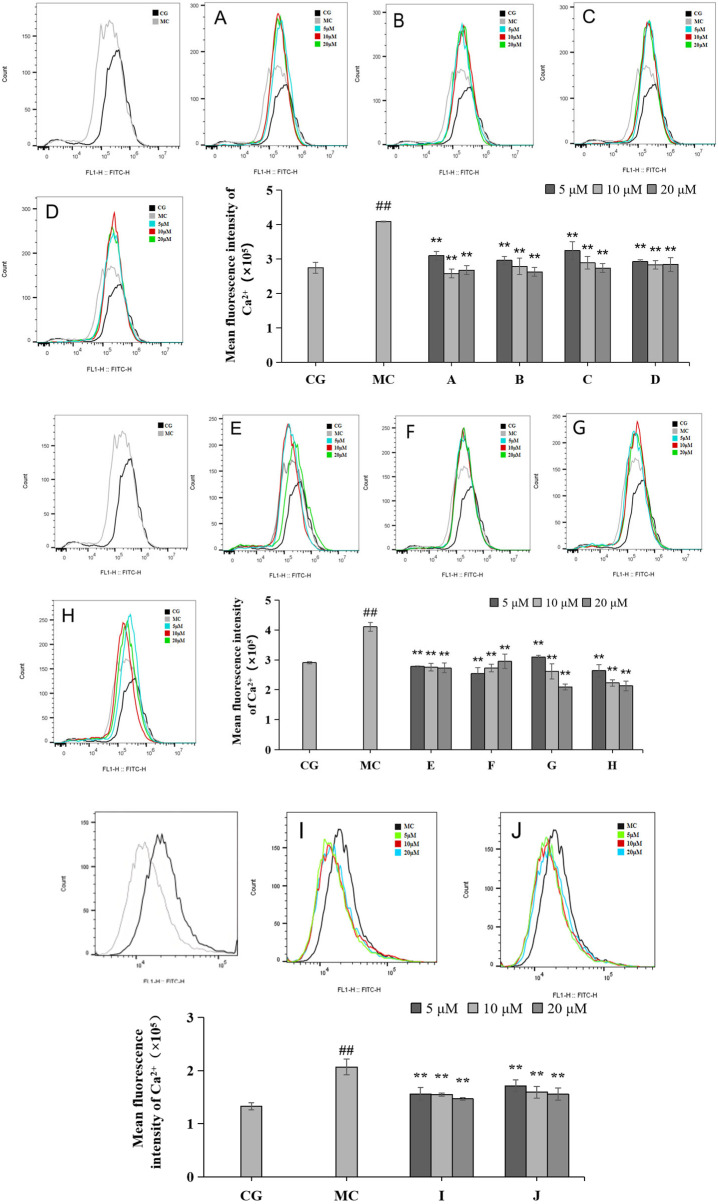
Effects of different concentrations of groups A–J on the Ca^2+^ levels measured via flow cytometry. Quantitative analysis of ROS fluorescence intensity (*n* = 3). The data are expressed as the mean ± standard deviation (SD) of three independent experiments. *## p* < 0.01 vs. control group (CG); ** *p* < 0.01 vs. model group (MC).

**Figure 7 molecules-28-07319-f007:**
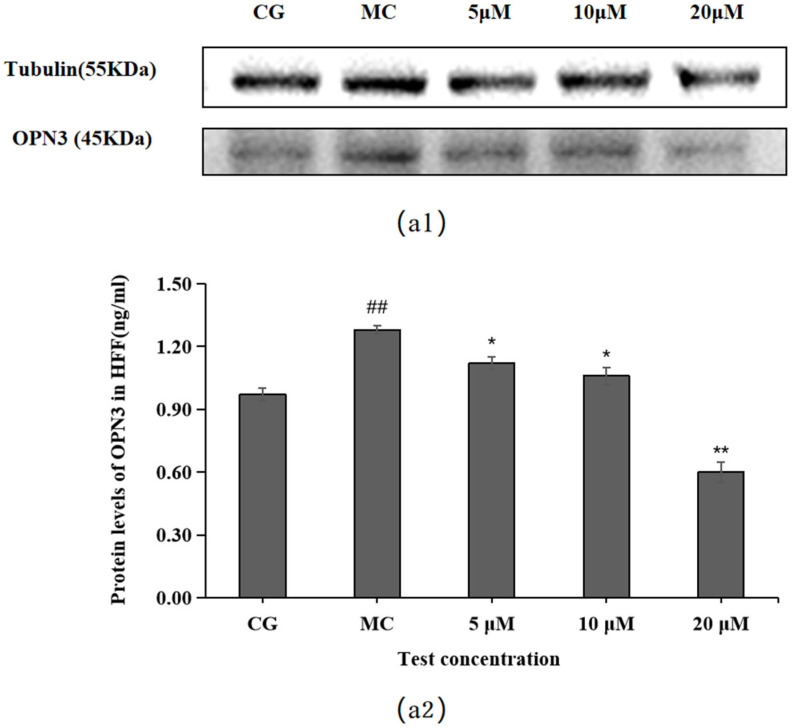
Effects of leontopodic acid A (I) on OPN3. (**a1**) MMP-1 expression was measured using a Western blotting assay; (**a2**) MMP-1 expression was visualized by ELISA. ## *p* < 0.01 vs. control group (CG); * *p* < 0.05 and ** *p* < 0.01 vs. model group (MC).

**Table 1 molecules-28-07319-t001:** Experimental groups of the samples.

Sample	Group	Sample	Group
Chlorogenic acid	A	Isochlorogenic acid	F
Isoquercitrin	B	Cynarin	G
Isochlorogenic acid A	C	Rutin	H
Cynaroside	D	Leontopodic acid A	I
Syringin	F	Leontopodic acid B	J

## Data Availability

Data is unavailable due to privacy.

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
