# Peer review of "Active Components of Leontopodium alpinum Callus Culture Extract for Blue Light Damage in Human Foreskin Fibroblasts"

_molecules, 2023, doi:10.3390/molecules28217319_

Round 1
Reviewer 1 Report
This research article presents the Effects of active components of Leontopodium alpinum callus 2 culture extract on blue light damage of human foreskin fibro-3 blasts. It appears to be an interesting topic. Overall, the manuscript is well-conceived, undertaken, written and presented. Minor revisions of content are required prior to acceptance for publication.
- Abstract:- Include the results of cell viability.
- Line 88: Add the citation of the previous study.
- Line 92: Add the cell model used.
- Line 96 (Figure 2): Redo the graph by presenting the concentration from 5 to 10 uM.
- Authors tested 10 active constituents of LACCE. But results in Figure 2 and Figure 3 only with 8 active constituents (A-H), I & J are missing. Please explain.
- Line 210-228: Suggest moving the contents to the introduction.
- Discussion: Discussion on the cell viability of 10 active constituents of LACCE is missing.
- Materials and Methods: Details such as the source of Leontopodium alpinum, sampling location, and procedure of sample processing and isolating the 10 active constituents are missing.
No major issues.
Author Response
|
Response to Reviewer Comments
|
||
|
1. Summary |
|
|
|
Thank you very much for taking the time to review this manuscript. Please find the detailed responses below and the corresponding revisions/corrections highlighted/in track changes in the re-submitted files.
|
||
|
2. Questions for General Evaluation |
Reviewer’s Evaluation |
Response and Revisions |
|
Does the introduction provide sufficient background and include all relevant references? |
Can be improved |
|
|
Are all the cited references relevant to the research? |
Yes |
|
|
Is the research design appropriate? |
Yes |
|
|
Are the methods adequately described? |
Must be improved |
|
|
Are the results clearly presented? |
Can be improved |
|
|
Are the conclusions supported by the results? |
Yes |
|
|
3. Point-by-point response to Comments and Suggestions for Authors |
||
|
Comments 1: Abstract:- Include the results of cell viability.
|
||
|
Response 1: We agree with this comment.Cell viability results have been added to the abstract. |
||
|
Comments 2: Line 88: Add the citation of the previous study. |
||
|
Response 2: Agree. I have added the citation of the previous studies.
|
||
|
Comments 3: Line 92: Add the cell model used. |
||
|
Response 3: Agree. I have added cell models used.
|
||
|
Comments 4: Line 96 (Figure 2): Redo the graph by presenting the concentration from 5 to 10 uM. |
||
|
Response 4: We have a little bit of a problem with this. The study in the article contains the concentration of 20uM. Is the graph less complete if it only shows 5, 10um concentrations?
|
||
|
Comments 5: Authors tested 10 active constituents of LACCE. But results in Figure 2 and Figure 3 only with 8 active constituents (A-H), I & J are missing. Please explain. |
||
|
Response 5: Agree. The graphs for active ingredients I and J are below A-H. The graphs have been reworked to make them clearer. |
||
|
Comments 6: Line 210-228: Suggest moving the contents to the introduction. |
||
|
Response 6: Agree. The content of lines 210-228 has been moved to the introduction. |
||
|
Comments 7: Discussion: Discussion on the cell viability of 10 active constituents of LACCE is missing. |
||
|
Response 7: Agree. We have added discussion of cell viability. |
||
|
Comments 7: Materials and Methods: Details such as the source of Leontopodium alpinum, sampling location, and procedure of sample processing and isolating the 10 active constituents are missing. |
||
|
Response 8: While one of our previous articles extracted and isolated the active ingredient in LACCE from acelbio (Effects and Mechanism of the Leontopodium alpinum Callus Culture Extract on Blue Light Damage in Human Foreskin Fibroblasts), this article examines the active ingredient's efficacy and the active ingredient is used by standards, not as a monomer isolated from the extract. The process of plant cell culture and extraction prior to LACCE involves company confidentiality and cannot be provided in the article. |
||
|
4. Responses to comments on the quality of English |
||
|
Comment 1: No major issues. |
||
|
Response 1: Thank you for your recognition. |
||
Reviewer 2 Report
Recommendation:
Include a discussion of the rationale for not testing the actives on control (non UV-light radiated cells). Clarify the rationale/basis for showing the data for actives A-J, and then two actives- was the selection for further studies with I & J actives based on the results of the A-J (all actives data).
Results:
2.1, Figure 2- edit the Y-axis to clarify the unit (%)- is it % of control © or % of the model group (NC). Include the names of the actives (A-J) in the legend or on the side of the figure
2.2, Figure 3- include in the legend and the figure the name of the active(s) in figures b1, b2. c1, c2
2.3, Figure 4- - include in the legend and the figure the name of the active(s) in figures e1, e2. f1, f2
2.6, Figure 7- edit the figure legend X and Y axis titles to specify what is tested and measured (with specific units).
In the figures- change the term expression to protein levels and the unit in bracket
Discussion- discuss all of the actives, were there significant differences in the effects of the different actives (A-J), and connect the effects and mechanisms
